# How Distributed Leadership Affects Social and Emotional Competence in Adolescents: The Chain Mediating Role of Student-Centered Instructional Practices and Teacher Self-Efficacy

**DOI:** 10.3390/bs14020133

**Published:** 2024-02-13

**Authors:** Zhenyu Li, Wei Liu, Qiong Li

**Affiliations:** 1Center for Teacher Education Research, Key Research Institute, Faculty of Education, Beijing Normal University, Beijing 100875, China; lizhyu@mail.bnu.edu.cn; 2School of Psychology, Guizhou Normal University, Guiyang 550025, China; liuwei@gznu.edu.cn; 3Taihang Branch of Center for Teacher Education Research of Beijing Normal University, Xingtai University, Xingtai 054000, China

**Keywords:** distributed leadership, student-centered instructional practices, teacher self-efficacy, adolescent social and emotional competence

## Abstract

The social and emotional competence of adolescents serves as the cornerstone for their success and future development. This study aims to explore the impact of distributed leadership on the social and emotional competence of adolescents, examining the mediating roles of student-centered teaching practices and teacher self-efficacy. Utilizing survey data from 7246 Chinese adolescents in the SESS project, the study employs a multi-level structural equation modeling approach for data analysis. The results indicate that distributed leadership positively predicts the social and emotional competence of adolescents. Furthermore, distributed leadership exerts indirect effects on adolescents’ social and emotional competence through the independent mediating roles of student-centered teaching practices and teacher self-efficacy, as well as a sequential mediation process involving student-centered teaching practices leading to teacher self-efficacy. This study elucidates how distributed leadership facilitates the development of adolescents’ social and emotional competence, confirming the supportive factors influencing these crucial capacities. Simultaneously, it provides valuable insights into the daily practices of teachers, principals, and administrators.

## 1. Introduction

In recent years, social–emotional competencies have gained recognition in global education as vital indicators for assessing students’ development and the quality of teaching [1]. These competencies encompass essential skills related to self-adaptation and social development that children acquire and apply [2]. They constitute a critical facet of students’ non-cognitive development [3], significantly impacting their academic progress and future success [4]. Research indicates that fostering social–emotional competence positively influences student achievement while mitigating negative behaviors and emotional distress [5,6]. Moreover, cultivating social–emotional competence during adolescence aids students in navigating future employment competition [7]. Studies have highlighted that individuals’ career paths and success in the job market are significantly influenced by their social–emotional competence [8]. This reflects the increasingly important trend of social–emotional competence in contemporary education. Hence, it becomes imperative to understand effective methods to enhance students’ social–emotional skills.

Distributed leadership, characterized by collaborative decision-making and coordinated action, has garnered attention in 21st-century education for bolstering school organizational capacity and supporting teacher growth [9,10]. Previous studies have demonstrated that distributed leadership contributes to the quality of teachers’ instructional methods and their adeptness in implementing teaching innovations [11,12]. Research also suggests that distributed leadership positively impacts teachers’ trust, motivation, organizational commitment, and self-efficacy [13,14,15]. This not only influences their instructional practices but also their social and emotional competencies. However, it remains unclear whether distributed leadership significantly contributes to students’ social and emotional competence. Additionally, existing research indicates that the influence of school leadership on student development is typically indirect [16]. Numerous researchers have extensively explored the correlation between school leadership and student academic achievement [17,18], leading to two prevailing conclusions: first, school leadership significantly influences student achievement. Second, the empirical association between school leadership and student achievement predominantly operates indirectly, mediated by various teacher and school-related factors [16,19].

Although previous studies have shown positive associations between distributed leadership and teacher self-efficacy [15], as well as student-centered instructional practices [11], there is limited understanding of whether and how these factors mediate the relationship between distributed leadership and students’ social–emotional competence.

This study utilizes data from the 2021 OECD Survey on Social and Emotional Skills (SSES) for Chinese adolescents. Its aim is to investigate how distributed leadership fosters adolescents’ social and emotional competence while elucidating the roles of student-centered instructional practices and teacher self-efficacy. The ultimate goal is to offer guidance and insights into enhancing adolescents’ social and emotional competence.

## 2. Literature Review

### 2.1. Social and Emotional Competence

The theoretical foundations of social and emotional competence trace back to earlier research exploring emotional intelligence, characterized by three primary frameworks: the Salovey and Mayer model [20], the Bar-On model [21], and the Goleman model [22]. These frameworks emphasize self-awareness, self-regulation, empathy, communication, and social interaction, significantly influencing subsequent definitions and assessment tools for social and emotional competence [23].

Among the numerous tools for assessing social and emotional competence, two frameworks stand out as particularly significant: the CASEL framework and the “Big Five Personality Traits” framework. The CASEL framework, proposed by the Collaborative for Academic, Social and Emotional Learning (CASEL), defines social–emotional learning (SEL) as the process by which children and adults acquire and effectively apply knowledge, attitudes, and skills required for understanding and managing emotions, setting and achieving positive goals, experiencing and expressing empathy towards others, establishing and maintaining positive relationships, and making responsible decisions [24]. According to this framework, social and emotional competence comprises five interlinked dimensions: self-awareness, self-management, social awareness, relational skills, and responsible decision-making. The CASEL framework is extensively referenced in SEL interventions and literature reviews [5].

Emerging research proposes an alternative definition of social and emotional competence within the Big Five framework [25]. This model encompasses Extraversion, Affinity, Dutifulness, Neuroticism, and Openness, offering a comprehensive descriptive categorization that consolidates various social and emotional competencies into a cohesive structure [26]. The SSES database established by OECD also adopts the Big Five personality model, dividing social and emotional competencies into five dimensions encompassing 15 sub-competencies. Notably, compared to the CASEL framework, the Big Five framework provides a more nuanced delineation of competencies and is better suited for extensive cross-sectional surveys [25]. Consequently, this study addresses research inquiries based on the dimensions constructed by the OECD.

### 2.2. Distributed Leadership and Social and Emotional Competence

Distributed leadership, initially proposed by Cecil Gibb in the Handbook of Social Psychology, gained traction after the nineties through continual development [27]. Various perspectives have shaped the conceptualization of distributed leadership, including situational learning [28], systems [29], process [30], and behavioral [31] viewpoints. Notably, the perspectives of Spillane and others hold considerable recognition among scholars. According to this theoretical perspective, distributed leadership constitutes the interaction among the leader, subordinates, and the situation, forming the foundation of leadership practice [32]. In this study, it specifically refers to an empowerment and shared responsibility management model within schools, authorizing teachers to engage in decision-making, fostering a favorable school climate [33]. Under this leadership model, teachers actively cultivate both individual and collective responsibility to address the diverse learning needs of their students [10].

So, how can the successful implementation of distributed leadership be achieved? This is an inherently crucial question. According to existing research, the success of schools in implementing distributed leadership depends on various factors, including organizational culture, the roles of leaders, technological support, and collaborative teamwork. Specifically, schools need to cultivate an organizational culture that supports innovation and distributed collaboration [34]. This involves providing appropriate resources and support to ensure that distributed teams can effectively cooperate. Additionally, school leaders need to possess effective communication and coordination skills to maintain connections within remote teams and to motivate and guide team members [35]. Furthermore, schools need to offer appropriate technological support and collaborative tools to facilitate effective communication and collaboration among distributed teams [36]. Moreover, a school culture conducive to the successful implementation of distributed leadership should emphasize collaboration and shared responsibility among leadership teams [37]. Finally, schools should provide training and professional development tailored for both faculty and leaders to enhance their capabilities in a distributed leadership environment [38]. These conditions represent key factors for the successful implementation of distributed leadership in schools, although specifics may vary based on the unique contexts of individual institutions.

Distributed leadership practices contribute significantly to fostering a supportive school culture, fostering positive student–teacher interactions, and nurturing robust student–teacher relationships—crucial components of adolescents’ social and emotional competence [39]. Moreover, implementing distributed leadership in instructional management positively influences students’ perceptions of teacher care, closely linked to empathy, self-awareness, and social awareness [40]. Additionally, distributed leadership correlates positively with individual creativity [41]. From this, it can be inferred that distributed leadership is closely related to the comprehensive development of students, especially certain sub-capabilities of social and emotional competence. However, whether distributed leadership directly influences adolescents’ social and emotional competence remains uncertain. Therefore, based on the above literature review and analysis, Hypothesis 1 is proposed in this study.

**Hypothesis** **1.***Distributed leadership significantly enhances adolescents’ social and emotional competence*.

### 2.3. The Mediating Role of Student-Centered Teaching Practices

Student-centered instructional practices, rooted in constructivism, advocate for learners actively constructing knowledge rather than passively receiving it [42]. These practices encompass diverse methods like problem-based learning, project-based learning, cooperative group learning, and inquiry-based learning [43]. Emphasizing student responsibility for their learning needs, this approach places the teacher in the role of a facilitator or organizer. Teachers within this framework must employ varied teaching methods flexibly to address diverse student needs, fostering continual improvement in students’ creativity, perseverance, organizational, interpersonal, and collaborative skills. Studies highlight that student-centered practices yield higher student scores compared to traditional methods, contributing significantly to students’ knowledge, skills, and qualities [44,45]. Specifically, they positively predict the use of deep learning methods and enhanced self-reported competencies, encompassing cognitive and practical skills, even in larger class settings [46].

Effective leadership styles and robust school support are vital for teachers to develop instructional skills emphasizing the significance of student-centered approaches. Distributed leadership, esteemed for its empowered and shared leadership concept, correlates positively with several school organizational factors influencing teacher-centered instructional practices, encompassing organizational change, teacher leadership, professional learning communities, teacher self-efficacy, and school climate [15]. Additionally, distributed leadership indirectly impacts teacher instructional practices through teacher collaboration and job satisfaction, where student-centered instructional practices constitute one of the sub-dimensions [11]. It can be inferred that the implementation of distributed leadership significantly influences teachers’ instructional activities, contributing to the advancement of student-centered teaching practices, ultimately enhancing the quality of teaching.

Furthermore, research has identified that teaching practices play a mediating role in the influence of the school climate on teachers’ self-efficacy, with a significant mediating effect [47]. This indicates that the impact of the school climate on teachers’ self-efficacy is largely mediated through teaching practices. However, a particular study points out that due to the limited influence of school leadership on student-centered teaching practices, student-centered teaching practices do not serve as a mediator in the impact of school leadership on teachers’ identity recognition [48]. From this, it can be inferred that the mediating role of student-centered teaching practices still warrants in-depth exploration. Additionally, it remains unclear whether student-centered teaching practices can act as a mediator in the impact of distributed leadership on the social and emotional competence of adolescents. Therefore, based on the aforementioned literature review, this study proposes Hypothesis 2:

**Hypothesis** **2a.***Distributed leadership significantly influences student-centered instructional practices*.

**Hypothesis** **2b.***Student-centered instructional practices significantly impact students’ social and emotional competence*.

**Hypothesis** **2c.***Student-centered instructional practices mediate the impact of distributed leadership on adolescents’ social and emotional competence*.

### 2.4. The Mediating Role of Teacher Self-Efficacy

Teacher self-efficacy embodies teachers’ beliefs in their ability to influence student learning outcomes, such as student interest and motivation, through their teaching activities [49]. It reflects teachers’ confidence in their teaching abilities. Empirical studies indicate a positive correlation between teacher self-efficacy and students’ perceived teacher emotional support [50], closely tied to adolescents’ social and emotional competence. Moreover, teacher efficacy significantly influences students’ perceived social relationships, enhancing their interpersonal skills and social awareness [51].

In various educational settings where teachers possess ample time for targeted instruction based on their discretion, teacher-led leadership models are prevalent [52]. Distributed leadership amplifies teacher self-efficacy through three primary channels [53]. First, by empowering teachers through distributed leadership practices, it cultivates a positive school climate that encourages knowledge sharing and collaborative efforts among teachers, thereby enhancing instruction [15]. Second, granting teachers greater control over their work environment motivates them to invest more in instruction preparation, implementation, and reflection [54]. Third, verbal encouragement and support from leaders play a pivotal role in boosting teachers’ self-efficacy [55]. Facilitating effective communication between administrators and teachers is essential in achieving shared goals.

Furthermore, several studies have established that teacher self-efficacy mediates the effects of distributed leadership on teacher-related aspects. For instance, it mediates the impact of distributed leadership on teachers’ job well-being and professional well-being [53], and indirectly influences students’ reading literacy [56]. Nonetheless, it remains unclear whether teacher self-efficacy also mediates the effects of distributed leadership on students’ social and emotional competence. Hence, based on this literature review, Hypothesis 3 is proposed:

**Hypothesis** **3a.***Distributed leadership significantly impacts teacher self-efficacy*.

**Hypothesis** **3b.***Teacher self-efficacy significantly influences adolescents’ social and emotional competence*.

**Hypothesis** **3c.***Teacher self-efficacy mediates the impact of distributed leadership on adolescents’ social and emotional competence*.

### 2.5. Student-Centered Teaching Practices and Teacher Self-Efficacy

A quasi-experimental study revealed the efficacy of project-based learning (PBL) in enhancing teacher self-efficacy [57]. It found that positive student responses to instructional practices could potentially mediate the relationship between PBL and teacher self-efficacy. Similarly, Holzberger et al. [58] conducted a longitudinal follow-up survey, illustrating that instructional practices significantly predicted teachers’ self-efficacy. Moreover, analyses using TALIS 2018 data confirmed the positive influence of instructional practices on teachers’ self-efficacy, a trend observed across diverse cultural contexts, including Chinese, Canadian, Finnish, Japanese, and Singaporean settings [59]. In light of the literature analysis, this study posits Hypothesis 4:

**Hypothesis** **4a.***Student-centered instructional practices positively impact teacher self-efficacy*.

**Hypothesis** **4b.***Student-centered instructional practices and teacher self-efficacy collectively mediate the effects of distributed leadership on adolescents’ social and emotional competence*.

## 3. Methods

### 3.1. Research Participants

This study utilized Chinese data from the 2019 OECD (Organization for Economic Cooperation and Development) Study on Social and Emotional Skills (SSES) among adolescents. The survey employed stratified sampling to gather data from primary and secondary school students, parents, teachers, and schools within the districts and counties under Suzhou City’s jurisdiction in China. It specifically focused on assessing adolescents’ social and emotional competence and examining the potential factors contributed by teachers, parents, and schools affecting this competence. Given the study’s emphasis on teacher variables impacting students’ social and emotional competence and the potential influence of social desirability bias on self-report measures, multiple sources of data were collected [3]. This included self-reports from students, reports from teachers, and reports from parents regarding adolescents’ social–emotional competence, ensuring data triangulation for improved accuracy. Data matching was conducted across the three groups—students, teachers, and parents—based on unique identifiers (student ID, teacher ID, and parent ID) within the survey data to facilitate subsequent data analysis. The adolescent sample for this study consisted of 7246 individuals, comprising 3409 girls (47.05%), 3824 boys (52.77%), and 13 (0.18%) with unspecified gender. The teacher sample comprises 3624 individuals, with 2511 being female teachers (69.29%) and 1111 being male teachers (30.66%). There are also 2 missing data points. The age of teachers ranges from 20 to 66 years, and their total teaching experience spans from 0 to 42 years. The parental sample comprises 7246 individuals. Among them, 4230 mothers completed the questionnaire, accounting for 58.38%, while 2642 fathers completed the questionnaire, constituting 36.46%. Additionally, 182 surveys were completed by other guardians, representing 2.51%. There are also 192 missing data points, making up 2.65% of the total. The age of guardians primarily falls within the range of 35 to 44 years.

### 3.2. Measures

#### 3.2.1. Distributed Leadership

In the SSES teacher questionnaire, the variable of distributed leadership was assessed using items aligned with those employed in the Teaching and Learning International Survey (TALIS) 2018 [60]. Given the study’s focus on the teacher-level distributed leadership variable, three items were designated for its measurement (see Appendix A), in accordance with existing research [61]. These items were rated on a Likert scale measuring the degree of agreement, ranging from 1 (strongly disagree) to 5 (strongly agree). A reliability and validity test was conducted on these three items, yielding a Cronbach’s alpha coefficient of 0.934. Results from the confirmatory factor analysis indicate that the standardized factor loadings for each item fall within the range of 0.845 to 0.948. It is noteworthy that, due to the saturated nature of the confirmatory factor analysis model with three items, specific fit information is not provided. Nevertheless, these findings affirm the robust reliability and validity of the distributed leadership variable.

#### 3.2.2. Student-Centered Teaching Practices

The SSES teacher questionnaire included the measurement of the variable of student-centered teaching practices [43,62,63], consisting of six items (see Appendix A). These items were measured on a four-point frequency scale ranging from 1 (almost never) to 4 (almost every lesson). A reliability and validity test was conducted on these six items, yielding a Cronbach’s alpha coefficient of 0.909. The results of the confirmatory factor analysis demonstrate a well-fitting model: X^2^ = 129.133, *df* = 8; CFI = 0.991; TLI = 0.983; RMSEA = 0.065; SRMR = 0.015. Simultaneously, the standardized factor loadings for each item range from 0.679 to 0.861, affirming the good reliability and validity of the student-centered teaching practices variable.

#### 3.2.3. Teacher Self-Efficacy

The SSES teacher questionnaire assessed the variable of teacher self-efficacy, utilizing items consistent with those employed in TALIS 2018 [60,62]. This variable was measured with seven items (see Appendix A) on a four-point frequency scale ranging from 1 (not at all) to 4 (a lot). A reliability and validity test was conducted on these seven items, resulting in a Cronbach’s alpha coefficient of 0.935. The results of the confirmatory factor analysis reveal a well-fitting model: X^2^ = 222.376, *df* = 10; CFI = 0.990; TLI = 0.978; RMSEA = 0.077; SRMR = 0.017. Additionally, the standardized factor loadings for each item range from 0.751 to 0.882, affirming the good reliability and validity of the teacher self-efficacy variable.

#### 3.2.4. Social and Emotional Competence

Drawing from the Big Five Model, the SSES evaluates this competency across five dimensions, each comprising three sub-competencies [62]. These dimensions include Task Competency (Responsibility, Perseverance, Self-Control), Emotional Regulation Competency (Resistance to Stress, Optimism, Emotional Control), Collaboration Competency (Empathy, Cooperation, Trust), Openness (Inclusiveness, Curiosity, Creativity), and Interaction Competency (Agreeableness, Boldness, Vigor).

In this study, adolescents’ self-assessed social and emotional competence, teacher-assessed social and emotional competence, and parent-assessed social and emotional competence were selected as dependent variables for analysis. Each subscale of the adolescents’ self-assessed and parent-rated social and emotional competence scales consisted of eight questions, measured on a scale of agreement (1: strongly disagree to 5: strongly agree). Meanwhile, each subscale of the teacher-evaluated social and emotional competence scale contained three questions, measured on a similar agreement scale. Due to the extensive number of items measuring adolescent social and emotional competence, this study refrains from presenting them here. Detailed information regarding the social and emotional competence of adolescents can be found in the SSES Technical Report [62].

The mean scores of the 15 sub-competencies were used to determine the adolescents’ social and emotional competence. Higher scores denoted greater social and emotional competence. These scores provided the final assessment for adolescents’ self-assessed, teacher-assessed, and parent-assessed social and emotional competence. The reliability of the assessments from adolescents, teachers, and parents regarding social and emotional competence has been thoroughly tested, yielding favorable overall results [64].

### 3.3. Data Analysis

Given the nested structure of the data utilized in this study, as illustrated in Figure 1, the social and emotional competence of adolescents represent variables at the student level, denoted as Level 1 variables. Conversely, distributed leadership, student-centered teaching practices, and teacher self-efficacy are variables at the teacher level, designated as Level 2 variables. Multiple students are nested within different teachers. Consequently, this study employs a Multilevel Structural Equation Modeling (MSEM) approach, utilizing Mplus 8.3 statistical analysis software for data analysis. The analytical process involves several steps:

Step 1: In the initial phase, we scrutinize the appropriateness and necessity of implementing a multilevel structural equation model. To achieve this, we assess the intra-class correlation coefficients (ICCs) for three dependent variables: adolescent self-assessed social and emotional competence, teacher-evaluated social and emotional competence, and parent-assessed social and emotional competence. Researchers have proposed two indicators for ICC, namely ICC-1 (ICC-1 = $\sigma_{b}^{2}$/($\sigma_{b}^{2}$ + $\sigma_{w}^{2}$)) and ICC-2 (ICC-2 = $\sigma_{b}^{2}$/($\sigma_{b}^{2}$ + $\sigma_{w}^{2}$/*n*)), where $\sigma_{b}^{2}$ is the between-cluster variance, $\sigma_{w}^{2}$ is the within-cluster variance and *n* is the average cluster size [65]. It is considered appropriate to proceed with multilevel analysis only when the value of ICC-1 exceeds 0.05, and the value of ICC-2 surpasses 0.50 [66].

Step 2: Descriptive statistics were employed to present comprehensive scores for distributed leadership, student-centered instructional practices, teacher self-efficacy, and adolescents’ social and emotional competence. Furthermore, this step assessed potential correlations among these variables.

Step 3: In this step, multilevel structural equation modeling is conducted to delve into the intricate relationships between variables. Specifically, addressing the limitations of null hypothesis testing, this study not only reports the results of hypothesis testing but also presents the effect sizes. Effect sizes serve as indicators measuring the strength of experimental effects or the intensity of associations between variables, and they are less influenced (or minimally influenced) by sample size [67]. To begin, drawing on previous research [68], this study employs the *f*^2^ statistic to calculate the effect size of independent variables on dependent variables (*f*^2^ = *R*^2^/(1 − *R*^2^), where *R*^2^ gauges the linear association strength between dependent and independent variables, representing the proportion of the dependent variable’s total variance explained by the independent variable). Cohen [68] proposed that *f*^2^ values of 0.02, 0.15, and 0.35 correspond to small, medium, and large effect sizes, respectively. Subsequently, for the effect size of the mediating effect, the most commonly used method involves calculating the proportion of the indirect effect to the total effect (*P_M_* = ab/c) [69]. Therefore, this study adopts this approach to compute the effect size of the mediating effect.

## 4. Results

### 4.1. Results of Intra-Class Correlation Coefficients

The results of the intra-class correlation coefficients are presented in Table 1. It is evident from the table that the ICC-1 value for adolescent self-assessed social and emotional competence is 0.06, with an ICC-2 value of 0.11. Although the ICC-1 value exceeds 0.05, the ICC-2 value is below 0.50. Consequently, utilizing adolescent self-assessed social and emotional competence as the dependent variable is deemed unsuitable for multilevel analysis. For teacher-evaluated social and emotional competence, the ICC-1 value is 0.42, and the ICC-2 value is 0.58. Therefore, employing teacher-evaluated social and emotional competence as the dependent variable is deemed appropriate for multilevel analysis. On the other hand, parent-assessed social and emotional competence exhibit an ICC-1 value of 0.01 and an ICC-2 value of 0.02. Hence, using parent-assessed social and emotional competence as the dependent variable is deemed unsuitable for multilevel analysis.

In summary, subsequent data analysis will exclusively include the analysis of teacher-evaluated social and emotional competence as the dependent variable. Analysis involving adolescent self-assessed social and emotional competence and parent-assessed social and emotional competence as dependent variables will be excluded.

### 4.2. Descriptive Statistics

The descriptive statistics presented in Table 2 indicate high scores for distributed leadership, student-centered instructional practices, teacher self-efficacy, as well as teacher-assessed social and emotional competence. The correlation analysis revealed a notable positive association among the variables.

### 4.3. Results of Structural Equation Modelling

Multilevel structural equation modeling was performed with teacher-assessed social and emotional competence as the dependent variable. The model fit indices were X^2^ = 1527.379, *df* = 114; CFI = 0.954; TLI = 0.945; RMSEA = 0.042; SRMR = 0.037 (Optimal values: CFI > 0.9; TLI > 0.9; RMSEA < 0.08; SRMR < 0.06 [70]). From the results of model fitting, it is evident that the model fits well, and the outcomes are compelling and convincing.

The results of the examination of standardized path coefficients are illustrated in Figure 1. From the results, it is evident that distributed leadership has a positive predictive effect on teacher-evaluated social and emotional competence (β = 0.183, *p* < 0.001), with an effect size of 0.100, approaching a medium effect size. Thus, Hypothesis 1 is confirmed. This implies that the implementation of distributed leadership contributes to the development of students’ social and emotional competence.

Distributed leadership significantly and positively influences student-centered teaching practices (β = 0.363, *p* < 0.001), with an effect size of 0.131, approaching a medium effect size. Therefore, Hypothesis 2a is confirmed. It indicates that comprehensive implementation of distributed leadership promotes teachers in implementing student-centered teaching practices, fostering student development.

Student-centered teaching practices positively predict teacher-evaluated social and emotional competence (β = 0.153, *p* < 0.001), with an effect size of 0.122, approaching a medium effect size. Hence, Hypothesis 2b is confirmed. This suggests that teachers engaging in student-centered teaching practices contribute to the cultivation and enhancement of students’ social and emotional competence.

Distributed leadership has a significant positive impact on teacher self-efficacy (β = 0.118, *p* < 0.001), with a relatively smaller effect size of 0.082. Thus, Hypothesis 3a is confirmed. This implies that the implementation of distributed leadership contributes to the improvement of teacher self-efficacy.

Teacher self-efficacy positively predicts teacher-evaluated social and emotional competence (β = 0.232, *p* < 0.001), with an effect size of 0.143, approaching a medium effect size. Therefore, Hypothesis 3b is confirmed. This indicates that the enhancement of teacher self-efficacy positively influences the social and emotional competence of adolescents.

Additionally, student-centered teaching practices significantly and positively influence teacher self-efficacy (β = 0.475, *p* < 0.001), with a substantial effect size of 0.328. Hence, Hypothesis 4a is confirmed. It suggests that engaging in student-centered teaching practices contributes to the enhancement of teacher self-efficacy.

The results of the examination of mediation effects among variables are presented in Table 3. It is evident from the results that student-centered teaching practices play a mediating role in the impact of distributed leadership on teacher-evaluated social and emotional competence (β = 0.055, *p* < 0.001). Additionally, the mediating effect size is 0.188 (*p* < 0.001). Therefore, Hypothesis 2c is confirmed. This indicates that distributed leadership not only directly influences the social and emotional competence of adolescents but also positively impacts these competencies indirectly through the implementation of student-centered teaching practices.

Similarly, teacher self-efficacy acts as a mediator in the influence of distributed leadership on teacher-evaluated social and emotional competence (β = 0.027, *p* < 0.001). The mediating effect size is 0.088 (*p* < 0.001). Thus, Hypothesis 3c is confirmed. This signifies that distributed leadership not only directly influences the social and emotional competence of adolescents but also indirectly enhances these competencies through the augmentation of teacher self-efficacy.

Furthermore, student-centered teaching practices and teacher self-efficacy jointly serve as a serial mediator in the impact of distributed leadership on teacher-evaluated social and emotional competence (β = 0.040, *p* < 0.001). The combined mediating effect size is 0.131 (*p* < 0.001). Consequently, Hypothesis 4b is confirmed. This suggests that distributed leadership not only directly influences the social and emotional competence of adolescents but also, through its impact on student-centered teaching practices and subsequent enhancement of teacher self-efficacy, ultimately fosters the social and emotional competence of adolescents.

Lastly, the total indirect effect of the entire mediation model is significant (β = 0.123, *p* < 0.001), with a total mediating effect size of 0.402 (*p* < 0.001).

## 5. Discussion

### 5.1. The Direct Impact of Distributed Leadership on Social and Emotional Competence

The research findings indicate that distributed leadership can positively predict teacher-evaluated social and emotional competence, with a moderate effect size. This suggests that the implementation of distributed leadership is conducive to the development of adolescents’ social and emotional competence to a certain extent. This leadership approach, emphasizing school empowerment and shared responsibilities among teachers, fosters an environment conducive to decision-making, ultimately benefiting student development [17]. The implementation of distributed leadership reflects a supportive school climate, encouraging positive social interactions between students and teachers. Consequently, adolescents learn crucial social and emotional skills like cooperation, empathy, helpfulness, trust, and tolerance.

### 5.2. The Mediating Role of Student-Centered Teaching Practices

The study reveals that student-centered teaching practices play a mediating role in the impact of distributed leadership on teacher-evaluated social and emotional competence, with a significant mediating effect size. Specifically, distributed leadership significantly and positively predicts student-centered teaching practices, with an effect size approaching moderate. Student-centered teaching practices, in turn, have a positive impact on teacher-evaluated social and emotional competence, with an effect size also approaching moderate. Therefore, implementing distributed leadership contributes to the better implementation of student-centered teaching practices by teachers, ultimately fostering adolescents’ social and emotional competence. This also indicates that the effective promotion of adolescents’ social and emotional competence through distributed leadership is, to some extent, mediated by student-centered teaching practices. By providing teachers with greater autonomy and encouraging innovative teaching methods such as cooperative group learning and project-based learning, distributed leadership enhances students’ critical thinking, problem-solving abilities, collaboration, and interpersonal skills. Various studies, including those by Bryk and Camburn, have demonstrated the positive impact of distributed leadership on improving teaching quality and student engagement [71,72]. Additionally, Shields’ research highlighted its role in aiding teachers’ decision-making and facilitating smooth instructional activities [73]. These studies collectively support the notion that implementing distributed leadership can foster innovative teaching approaches, enhance teaching quality, and contribute to the holistic development of adolescents.

In June 2019, the State Council emphasized the significance of contextualized teaching and project-based learning in enhancing the quality of compulsory education. Adolescents’ social and emotional competence encompasses multiple dimensions, where authentic problem scenarios serve as ideal contexts to assess and promote skills like creativity, emotional regulation, authentic problem-solving, and collaboration. However, implementing these student-centered instructional practices depends on the autonomy and decision-making authority granted to teachers by distributed leadership. Consequently, distributed leadership plays a crucial role in encouraging diverse models of student-centered instructional practices, ultimately enhancing students’ social and emotional competence. Aligning with existing research and educational policies, these findings underscore the indirect yet positive predictive influence of distributed leadership on adolescents’ social and emotional competence by shaping student-centered instructional practices.

### 5.3. The Mediating Role of Teacher Self-Efficacy

From the results, it is evident that teachers’ self-efficacy plays a mediating role in the impact of distributed leadership on teacher-evaluated social and emotional competence, with a significant mediating effect size. Specifically, distributed leadership significantly and positively predicts teachers’ self-efficacy, with an effect size not far from moderate. Teachers’ self-efficacy positively influences teacher-evaluated social and emotional competence, with an effect size approaching moderate. Therefore, the implementation of distributed leadership enhances teachers’ self-efficacy, and the improvement in teachers’ self-efficacy ultimately contributes to the development of adolescents’ social and emotional competence. This also illustrates that the effective promotion of adolescents’ social and emotional competence through distributed leadership is, to some extent, mediated by teachers’ self-efficacy. As an emerging model in school management, distributed leadership empowers teachers with decision-making authority, fostering a positive environment that stimulates their motivation, enhances their self-efficacy, and fosters a sense of belonging. Operating within this leadership model encourages teachers to continuously improve their teaching competence, thereby promoting adolescents’ social and emotional development.

Muijs’ study identified a positive correlation between distributed leadership, teacher self-efficacy, and teacher engagement [74]. It emphasized that implementing distributed leadership signals support, trust, and affirmation to teachers, nurturing their role identity and enhancing their self-efficacy [75]. Increased involvement in school decision-making processes also contributes to teachers’ confidence, self-esteem, and belief in their ability to positively impact the overall development of young individuals. Proponents of distributed leadership stress its potential to establish supportive school environments where teachers collectively take responsibility to cater to diverse student learning needs [76].

### 5.4. Chain Mediation of Student-Centered Teaching Practices and Teacher Self-Efficacy

The research findings indicate that student-centered teaching practices and teachers’ self-efficacy jointly play a sequential mediating role in the impact of distributed leadership on teacher-evaluated social and emotional competence, with a significant mediating effect size. Specifically, student-centered teaching practices significantly and positively affect teachers’ self-efficacy, with a substantial effect size. This indicates that distributed leadership not only promotes adolescents’ social and emotional competence through the separate mediating effects of student-centered teaching practices and teachers’ self-efficacy but also, to some extent, influences adolescents’ social and emotional competence through the sequential mediating effects of student-centered teaching practices and teachers’ self-efficacy. There is a potential association among distributed leadership, student-centered instructional practices, teacher self-efficacy, and adolescents’ social and emotional competence, as highlighted in the literature analysis. This study further delineates the mechanisms between these variables, elucidates the combined effect of distributed leadership on adolescents’ social and emotional competence, and presents a new avenue for influencing it.

In line with social exchange theory, which explores reciprocity norms within organizations, there are concepts of both direct and indirect reciprocity [77]. Direct reciprocity involves mutual help where the recipient returns the favor to the giver. Indirect reciprocity encompasses a third party, often within the organization, creating a complex pattern of social interaction. In this study, employing the principle of indirect reciprocity, the school supported teachers by implementing distributed leadership, benefiting both the teachers and the organization. Empowering teachers and promoting teamwork and leadership cultivated teachers’ autonomy, resulting in improvements in curriculum content and diverse student-centered teaching approaches.

As a consequence, teachers empowered students by adopting varied student-centered practices, encouraging student initiative within the classroom. Through this process, teachers developed deeper classroom insights, increased confidence in teaching, and enhanced their self-efficacy, reaping direct benefits from the implemented strategies. Ultimately, students, as third-party beneficiaries, received valuable feedback from both the school and teachers, contributing to the well-rounded development of their social and emotional competence.

Drawing upon the analysis presented above, this study’s distinctive contribution lies in quantitatively demonstrating the positive impact of distributed leadership on the social and emotional competence of adolescents. While previous research has explored the significant role of school leadership in student development, the influence of distributed leadership on students’ social and emotional competence has remained unclear. Furthermore, this study delves into the underlying mechanisms through which distributed leadership affects the social and emotional competence of adolescents, providing novel insights for subsequent research. This complex relationship has not been thoroughly established in previous studies. Specifically, by constructing a sequential mediation model involving student-centered teaching practices and teachers’ self-efficacy, we elucidate how distributed leadership influences the social and emotional competence of adolescents, offering a fresh perspective for research in this area.

In summary, the findings of this study furnish crucial information on how to enhance the social and emotional competence of adolescents. It distinctly highlights the pivotal roles of distributed leadership, student-centered teaching practices, and teachers’ self-efficacy in the development of students.

## 6. Implications

First, empowering teachers and facilitating shared decision-making are pivotal in implementing distributed leadership. Centralizing power in the school principal’s hands can lead to drawbacks, making empowerment and sharing of responsibilities critical to the principal’s success [78]. School decision-making is intricate, often spanning various subjects, reflecting specialization’s complexity. Principals, unable to possess all expertise, depend on teachers from diverse disciplines to collaborate, necessitating the delegation of leadership authority. Simultaneously, nurturing teachers’ leadership abilities is vital. Teachers need decision-making prowess to underpin informed school choices. Hence, regular training is essential to enhance teachers’ decision-making skills.

Second, effective social and emotional competence cultivation should be integrated into daily classroom teaching. Encouraging student-centered learning environments through group cooperative, problem-oriented, and project-based approaches empowers students in the classroom. Teachers should promote group activities, fostering adolescents’ responsibility, empathy, cooperation, trust, communication, and problem-solving skills. Guiding students actively, offering feedback, trusting their capabilities, nurturing divergent thinking, and encouraging innovation contribute to stimulating their potential.

Lastly, bolstering teachers’ confidence and beliefs in their teaching process is crucial for enhancing their self-efficacy. According to Bandura, the external environment significantly influences self-efficacy [79]. School leaders play a key role in creating a supportive climate, offering timely guidance, and providing external support to boost teachers’ self-efficacy. Diversified training opportunities are pivotal for deeper professional knowledge and skills mastery, reinforcing professionalism. Encouraging teachers to develop unique curricula and use diverse teaching methods bolsters their confidence in handling their work, thereby enhancing their sense of achievement and value. Teachers themselves should actively build their self-efficacy by recording successful teaching experiences, comparing teaching outcomes, and engaging in professional development through collaboration and exchange with peers.

## 7. Limitations

The study’s conclusions were drawn through quantitative research methods utilizing relatively homogeneous data. To further substantiate and broaden these findings, future research could incorporate qualitative or mixed methods for data collection.

Moreover, this study employed cross-sectional data, constructing relationships between variables based on existing theories and prior literature. Yet, the determination of causal relationships remains incomplete. Future research employing longitudinal tracking could delve deeper into the causal links and dynamic characteristics among distributed leadership, student-centered teaching practices, teacher self-efficacy, and social and emotional competence.

Lastly, potential additional mediating pathways between distributed leadership and students’ social–emotional competence warrant exploration in future research endeavors.

## 8. Conclusions

The study aimed to explore the connections between distributed leadership, student-centered instructional practices, teacher self-efficacy, and students’ social–emotional competence. It revealed that distributed leadership, student-centered instructional practices, and teacher self-efficacy significantly impact students’ social–emotional competence. Additionally, the study reveals that distributed leadership exerts indirect effects on adolescents’ social and emotional competence through the independent mediating roles of student-centered teaching practices and teacher self-efficacy, as well as a sequential mediation process involving student-centered teaching practices leading to teacher self-efficacy. The research not only provides robust support for understanding how distributed leadership fosters the development of adolescents’ social and emotional competence but also offers valuable guidance and insights for enhancing and elevating the social and emotional competence of young individuals.

## Figures and Tables

**Figure 1 behavsci-14-00133-f001:**
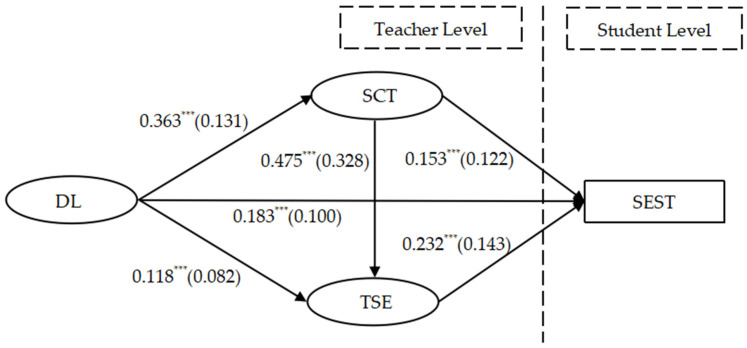
Chain mediation model of DL and SEST. Note: The values outside the parentheses represent estimates, while those inside represent the effect size, denoted by *f*^2^. *** *p* < 0.001.

**Table 1 behavsci-14-00133-t001:** Results of intra-class correlation coefficients.

Variables	Index	Estimated Value
Adolescents’ self-assessed social and emotional competence	ICC-1	0.06
	ICC-2	0.11
Teacher-assessed social and emotional competence	ICC-1	0.42
	ICC-2	0.58
Parent-assessed social and emotional competence	ICC-1	0.01
	ICC-2	0.02

**Table 2 behavsci-14-00133-t002:** Descriptive and correlation analysis.

Variables	M	SD	1	2	3	4
1. DL	3.95	0.91	1			
2. SCT	3.03	0.68	0.36 **	1		
3. TSE	3.47	0.59	0.29 **	0.52 **	1	
4. SEST	3.54	0.32	0.31 **	0.34 **	0.36 **	1

Notes: ** *p* < 0.01; DL = distributed leadership; SCT = student-centered teaching practices; TSE = teacher self-efficacy; SEST = social and emotional competencies of teacher evaluation.

**Table 3 behavsci-14-00133-t003:** Indirect effects and total indirect effects test results.

Type of Effect	β	SE	95% Confidence Interval	Effect Sizes
DL→SCT→SEST	0.055 ***	0.010	[0.035, 0.076]	0.180 ***
DL→TSE→SEST	0.027 ***	0.005	[0.018, 0.037]	0.088 ***
DL→SCT→TSE→SEST	0.040 ***	0.005	[0.030, 0.050]	0.131 ***
DL→SEST (total indirect effects)	0.123 ***	0.010	[0.102, 0.143]	0.402 ***

Notes: *** *p* < 0.001. DL = distributed leadership; SCT = student-centered teaching practices; TSE = teacher self-efficacy; SEST = social and emotional competencies of teacher evaluation.

## Data Availability

The study used data from the SSES (https://www.oecd.org/education/ceri/social-emotional-skills-study/data.htm (accessed on 4 December 2023)).

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
