# Peer review of "How Distributed Leadership Affects Social and Emotional Competence in Adolescents: The Chain Mediating Role of Student-Centered Instructional Practices and Teacher Self-Efficacy"

_behavsci, 2024, doi:10.3390/bs14020133_

Round 1

Reviewer 1 Report

Comments and Suggestions for Authors

The article aims to investigate how distributed leadership fosters adolescents' social and emotional competence, building on data from the 2021 OECD Survey on Social and Emotional Skills (SSES) for Chinese adolescents. It also states that ”the ultimate goal is to offer guidance and insights into enhancing adolescents' social and emotional skills”, providing insights for the educational actors. 

The article is well structured, clearly written, and well articulated in the literature. 

Some revisions are suggested, in relation with the following aspects:

Clarifications are to be made in the methodology part, mainly with relation to the sampling. There are also mentioned the parents; even their role is not described in the theoretical part. Also, the article deals with the views of the teachers, while the sampling mentioned only the boys and girls.

Some more clarifications are needed in the discussion part in relation to the second aim of the study, also mentioned in the abstract. It states that it provides valuable data for every day practices of school administrators. Can this be clarified?  

We would like to see a little bit more elaborated comments while presenting the results, to reflected them more friendly for the reader. 

We would like to see in the discussion and the conclusion part highlighted in a better way what an international reader should take with him/her from the article and how the data adds to the international debate. 

I hope my suggestions can help you enhance a well-articulated article. 

Reviewer 2 Report

Comments and Suggestions for Authors

Main points: It does not appear that Distributed Leadership was actually measured despite the claims of the paper. Very serious concerns for the validity of the paper overall because of this. Other measures were also not clearly outlined as accurate and valid measures of the constructs for which they are purported to measure. This is unfortunate as the paper held a lot of promise. More detailed comments are in the attached document.

Reviewer 3 Report

Comments and Suggestions for Authors

The suggested study aims to answer the question:   How distributed leadership affects social and emotional competence in adolescents and elucidate mediating factors as possible contributors to the social and emotional competence in adolescents. The writing is impressive in clarity and register, and the literature review is encompassing, though shortening it could assist in focusing the reader on the issue at hand. Some parts of the introduction and literature review are too informative rather than critically edited. The rationale for the hypotheses precedes them, yet hypothesis 2c would merit further justification. The explanations for the results that appear in the discussion section could benefit from a theoretical conceptualization of the reciprocal ties that were described, such as the ecological approach and its interpretation in teacher leadership. 
